# Inflammation-Associated Cytotoxic Agents in Tumorigenesis

**DOI:** 10.3390/cancers16010081

**Published:** 2023-12-22

**Authors:** Jürgen Arnhold

**Affiliations:** Institute of Medical Physics and Biophysics, Medical Faculty, Leipzig University, Härtelstr. 16-18, 04107 Leipzig, Germany; juergen.arnhold@medizin.uni-leipzig.de

**Keywords:** cytotoxic agents, antagonizing principles, chronic inflammation, hypoxia, tumor cells, tumor microenvironment, immunosuppression, redox homeostasis, matrix remodeling

## Abstract

**Simple Summary:**

This review summarizes the present knowledge of cytotoxic agents and protective systems against these damaging substances in tumors. In tumor cells, enhanced levels of cytotoxic agents are usually counteracted by an overexpression of protective mechanisms. In this manner, tumor cells can even survive therapeutically induced stress situations. Tumor cells also affect immune cells and other cells in their close neighborhood in such a way that cells in this tumor microenvironment change their properties and promote tumor progression. Numerous examples are given for how the disturbed balance between inflammation-associated cytotoxic agents and antagonizing principles is related to tumor growth, tumor cell invasion, and metastasis. A thorough knowledge of these mechanisms is mandatory for the implementation of novel therapeutic approaches against cancers.

**Abstract:**

Chronic inflammatory processes are related to all stages of tumorigenesis. As inflammation is closely associated with the activation and release of different cytotoxic agents, the interplay between cytotoxic agents and antagonizing principles is highlighted in this review to address the question of how tumor cells overcome the enhanced values of cytotoxic agents in tumors. In tumor cells, the enhanced formation of mitochondrial-derived reactive species and elevated values of iron ions and free heme are antagonized by an overexpression of enzymes and proteins, contributing to the antioxidative defense and maintenance of redox homeostasis. Through these mechanisms, tumor cells can even survive additional stress caused by radio- and chemotherapy. Through the secretion of active agents from tumor cells, immune cells are suppressed in the tumor microenvironment and an enhanced formation of extracellular matrix components is induced. Different oxidant- and protease-based cytotoxic agents are involved in tumor-mediated immunosuppression, tumor growth, tumor cell invasion, and metastasis. Considering the special metabolic conditions in tumors, the main focus here was directed on the disturbed balance between the cytotoxic agents and protective mechanisms in late-stage tumors. This knowledge is mandatory for the implementation of novel anti-cancerous therapeutic approaches.

## 1. Introduction

Inflammation is related to all stages of tumorigenesis, such as the initiation, promotion, progression, and metastasis [1,2,3]. Inflammatory processes are primarily directed to eliminate by means of the immune systems and the activation of components in the acute phase, complement, coagulation, and contact systems any harm from the host that can disturb normal functioning and tissue homeostasis [4,5].

In tumors, the associated inflammatory process is persistent [6]. During tumor development and the interaction between immune and tumor cells, three major phases can be distinguished, according to the concept of immunoediting [3,7]. These phases are schematically presented in Figure 1. The first phase, the elimination phase, corresponds to the original concept of immune surveillance, suggesting that immune cells are able to eliminate transformed cells [8,9]. Natural killer (NK) cells and cytotoxic T lymphocytes (CTLs) are the key players in the recognition and killing of transformed cells [10,11]. It is generally assumed that the formation and elimination of unwanted transformed cells occurs often in our organism.

During the second phase, the equilibrium phase, the elimination of tumors cells and the development of tumor-mediated immune suppressing strategies co-exist. Thus, the elimination of unwanted cells is impeded, and part of these cells can survive. In this phase, tumor immunogenicity is reduced, and a pro-inflammatory tumor microenvironment (TME) is generated that promotes immune tolerance [3].

These alterations in the immune response are gradually strengthened to the immune escape that is characteristic of the third phase, the escape phase. As a result, the immune response is manipulated in multiple ways by tumor cells and the TME, ensuring that tumor growth and metastasis can occur and the processes of elimination of tumor cells by the host’s immune system are slowed down. The progressive development of tumor-mediated immune escape is closely associated with major changes in the tumor cell properties and conditions in the TME, such as increasing the deficiency of dioxygen, the accumulation of lactate, altered routes in the supply of energy substrates, the activation of stress-related genes and oncogenes, nutrient depletion, reversed pH gradient on tumor cells, and matrix remodeling. In general, these altered conditions promote the survival of tumor cells, favor the suppression of immune defense reactions against tumor components, elevate proliferative activities, and contribute to tumor progression [3,12].

During the gradual development of tumors, many immunological relevant factors are converted from an anti-tumor behavior in the early stages to tumor-promoting properties in advanced tumors. Concurrently, many therapeutic procedures are increasingly restricted in their efficiency during tumor progression and fail in the late stages. In cancer research, the focus is, therefore, directed to study and identify the immunosuppressive mechanisms in late-stage tumors with the hope of finding novel ways for therapeutical intervention [12,13,14,15].

In tumor-associated inflammations, host-derived cytotoxic agents originate from activated immune cells, damaged tissue cells in the TME, dysfunctional cellular processes in tumor cells, and can additionally result from therapeutic applications like radio- and chemotherapy. To prevent the disastrous action of these cytotoxic agents, numerous ready-to-use protective mechanisms exist that neutralize these cytotoxic agents. Disturbances in the control of host-derived cytotoxic agents by antagonizing principles are associated with the development of chronic inflammatory disease states [16].

In this review, the knowledge of chronic inflammatory processes during tumorigenesis is highlighted. Special focus is directed on the interplay between host-derived cytotoxic agents and antagonizing principles in late-stage tumors and how alterations in this balance contribute to the survival of tumor cells and tumor progression.

## 2. Chronic Inflammatory States and Altered Conditions in Tumors

### 2.1. Immunosuppression during Acute and Chronic Inflammation

Regarding the time elapsed, an acute inflammation can be subdivided into two major phases with distinct differences in the mediator profile and immune cell functions. First, in response to pathogen-associated molecular patterns (PAMPs) and damage-associated molecular patterns (DAMPs), pro-inflammatory cascades are induced via the activation of pattern recognition receptors (PRRs) [17,18]. As a result, immune cells are recruited and activated onto inflammatory sites and the maturation of dendritic cells is induced for the presentation of antigens to T cells [19]. Typical pro-inflammatory mediators are cytokines, such as interleukin-1β (IL-1β), interleukin-6 (IL-6), and tumor necrosis factor α (TNF-α), and the acute-phase proteins C-reactive protein (CRP) and serum amyloid A (SAA) [4,5,20,21,22]. These activities are directed to deactivate pathogens, eliminate virus-infected or transformed cells, remove any damaged cell material, and create better conditions for ongoing immune reactions. This very common concept of pattern recognition is valid for the initiation of novel inflammatory events in a broad range of diseases, including infections, rheumatoid arthritis, atherosclerosis, Alzheimer’s disease, and many others, as well as tumors.

During inflammatory response, different host-derived cytotoxic agents are released or generated, which are derived from activated immune and undergoing tissue cells, and in tumors also from tumor cells. These agents generally execute a dual role in living tissues. They are an essential part in many physiological functions, especially in immune cell-triggered processes. Cytotoxic agents are mandatory for combatting microbes, to restore tissue homeostasis after any threat, and to ensure normal functioning of the physiological processes in the organism [4]. Otherwise, they can damage unperturbed tissues and contribute to the initiation of novel inflammatory cascades. According to their mode of action, cytotoxic agents can be differentiated into oxidant-based agents (reactive species, oxidized heme proteins, free heme, transition metal ions) and protease-based agents (serine proteases, matrix metalloproteases, pro-inflammatory peptides) [16]. In healthy tissues, cytotoxic agents are inactivated by already existing antagonizing principles.

In the second phase of an acute inflammation, the termination or resolution phase, destroyed cells and tissues are replaced by novel synthesized material via the induction of proliferative processes. The pro-inflammatory activities of immune cells are downregulated, and finally the former homeostasis is restored. The typical transiently acting cytokines of this phase are transforming growth factor β (TGF-β) and IL-10 [23,24,25]. In addition, growth factors [26,27] and lipid mediators, such as lipoxins and resolvins, are involved in the resolution of inflammation [28,29]. Cell signaling is characterized by the induction of the STAT3 pathway. Macrophages are polarized from the M1 type to M2 type [30,31]. The accumulation of myeloid-derived suppressor cells (MDSCs) contributes to the development of a transient immunosuppression in inflamed areas [32,33]. MDSCs are immature immune cells that promote immunosuppressive effects on lymphocytes, natural killer cells, macrophages, and dendritic cells [34]. In general, the aforementioned mechanisms impede the hyperactivation of pro-inflammatory cells and prevent any excessive tissue damage by host-derived cytotoxic agents during the resolution of inflammation. After the resolution, the number of immune cells and mediators is reduced to a level typical of healthy tissues [35]. Cytotoxic agents and their antagonizing counterparts are also involved in the resolution of inflammation, especially in the processes of matrix remodeling.

Under chronic inflammatory conditions, inflammation is only insufficiently terminated as pro-inflammatory cascades are activated again and again due to the incomplete inactivation of host-derived cytotoxic agents and the continuing release of DAMPs and antigens. At chronic inflammatory sites, pro-inflammatory mediators, the recruitment of immune cells, and tissue damage co-exist with a plethora of immunosuppressive mechanisms, such as the prevalence of MDSCs, the presence of inflammation-resolving and proliferative factors, as well as the depletion of essential metabolites for appropriate immune defense [16]. The resulting long-lasting immunosuppression can seriously affect the health status and enhance susceptibility to chronic infections and comorbidities [36,37]. For example, in elderly persons, persistent inflammation is closely associated with immune dysfunction. This condition is known as inflammaging [38], which is regarded as a risk factor for life-threatening diseases and adverse health outcomes [39,40]. In general, pro-inflammatory mechanisms and immunosuppressive conditions are closely associated with each other in many chronic diseases, including cancer.

In chronic inflammations, serious health problems can arise from the excessive release or generation of cytotoxic agents and from the decline or exhaustion of protective mechanisms. In consequence, the following release of novel DAMPs, antigens, and cytotoxic agents promotes ongoing inflammatory events and prevents the termination of inflammatory cascades. The worst case is a very low capacity of antagonizing principles or their exhaustion with the consequence of septic complications and organ failure [16]. This concept of the incomplete inactivation of cytotoxic agents by protective principles explains under which conditions an inflammation becomes persistent. It also provides the basis for a better understanding of the underlying molecular mechanisms for a wide range of chronic inflammatory diseases, including cancer.

In the description of the molecular processes during tumorigenesis, numerous cytotoxic agents derived from tumor cells, immune cells, and other tumor-associated cells are more expressed and exhibit higher activities compared to healthy tissue areas. In general, these agents are able to damage transformed cells with the subsequent elimination of these cells by the immune system. This is mostly observed during the early phases of tumorigenesis. However, in response to stress and the increasing accumulation of cytotoxic agents, numerous antagonizing principles are markedly upregulated in advanced tumors and promote the survival of tumor cells and protection against stress induced by chemo- and radiotherapy.

In Section 3 and Section 4, details about the interplay between cytotoxic agents and the protective mechanisms will be given concerning tumor development. As the action of cytotoxic agents highly depends on the altered conditions in tumor cells and the TME, Section 2.2 summarizes the key processes of immunosuppression and metabolic reprogramming occurring during tumorigenesis.

### 2.2. Key Elements of Immunosuppression in Tumors

Tumor progression and chronic inflammatory processes are highly linked to each other [1,2,3,6]. In the creation of an immunosuppressive milieu, both tumor cells and the TME contribute multiple mechanisms to the manipulation of the host’s immune answer and to tissue remodeling in tumors. An overview of the major mechanisms of tumor cell-mediated processes in the TME and immunosuppression is given in Figure 2.

During tumor progression, the principal processes involved in immunosuppression and tissue remodeling during the resolution phase of an acute inflammation are also active. Whereas these mechanisms are only transiently expressed in a subsiding inflammation, they act permanently and more intensively in tumors. In cancer cells, several protective systems are highly upregulated in response to a stress-mediated increase of cytotoxic agents (see Section 3). Further, these cells actively secrete inflammation-resolving molecules (TGF-β, IL-10) and exosomes that affect the conditions and properties of immune cells in the TME [41]. Importantly, natural killer cells (NK) and cytotoxic T cells, which are able to deactivate tumor cells, are suppressed in their activity by the release of TGF-β [42,43,44]. In the TME, tumor-associated macrophages (TAMs) are triggered from the pro-inflammatory M1 type to the immune-resolving M2 type [45]. Regulatory T cells (Treg) and various types of myeloid-derived suppressor cells (MDSCs) accumulate in the TME during tumor progression. These cell types suppress the activation of other immune cells and contribute to the survival of tumors [3,13,32,35,41,46,47,48]. With their cargo, cancer cell-derived exosomes are able to modulate different processes of tumor progression such as angiogenesis, metastasis, and survival [41].

MDSCs are formed as all other myeloid cells from precursors in the bone marrow. Under inflammatory conditions, the need for novel immune cells increases and these cells undergo a forced formation process, known as emergency myelopoiesis, leading to immature properties. In most kinds of cancers, the accumulation of granulocytic MDSCs (G-MDSCs) dominates over monocytic MDSCs (M-MDSCs) [49]. In glioma, M-MDSCs are more expressed than G-MDSCs [49]. In humans, a third, less pronounced type of MDSCs, early-stage MDSCs (e-MDSCs), has been described [50,51]. The MDSC subtypes are usually identified by their morphological properties, density, and surface markers [49].

In the TME, immunosuppressive conditions can be differentiated by their underlying mechanisms as enzyme-dependent, cytokine-dependent, and immune checkpoint-dependent [14]. In addition, the expression of neoantigens and antigen-presenting molecules like MHC-1 are decreased in cancer cells [52,53].

Enzyme-dependent immunosuppressive pathways are characterized by a depletion of essential metabolites, such as ATP and tryptophan, or by an enhanced formation of metabolites like adenosine, citrulline, kynurenine, and prostaglandin E2 [14,54,55]. Several proteins (for example ectonucleotidases CD39 and CD73 [56] and indoleamine 2,3-dioxygenase [57]), which are involved in these pathways, are highly upregulated in tumors.

Cytokine-dependent immunosuppression can be induced by the activation of the IL-6/STAT-3 pathway that inhibits DC maturation and activates effector T cells [58,59]. Vascular endothelial growth factor A (VEGF-A) is immunosuppressive by the inhibition of the DC functions and maturation, by the infiltration of immunosuppressive cells (Treg, MDSC, and TAMs) into tumors, by reducing cytotoxic CD8^+^ T cell infiltration into tumors, and by the expression of the factors involved in the exhaustion of these cells [60,61,62]. Cytokines of the IL-10 and TGF-β pathways activate Treg and promote immunosuppression [63].

Several immune checkpoints are known to be involved in immunosuppressive activities. The inactivation of T cells is possible via the binding of a specific ligand to the inhibitory T cell receptor programmed cell death protein 1 (PD-1) [14]. This ligand known as PD-L1 or CD274 is expressed on the surface of a tumor and various other cells. The overexpression of PD-L1 on tumor cells is associated with a poor prognosis and the evasion of T cell recognition of cancers [64]. The PD-1/PD-L1 axis represents an important element in the generation of an immunosuppressive TME [14]. Other inhibitory immune checkpoint molecules on T cells are cytotoxic T lymphocyte-associated protein 4 (also known as CD 152), LAG3, TIM-3, and the tyrosine-based inhibitory domain (TIGIT) [65,66,67]. Cancer cells can express TIGIT ligands, such as CD155, and overcome cancer immunity.

In growing and metastasizing tumors, the balance between tumor-promoting and tumor-destroying factors is shifted towards the first types of factors. Although, many features of tumor-induced immunosuppression are known, this knowledge is incomplete concerning the interrelations between them and the consequences for further tumor processing. Numerous anti-tumor therapies, such as targeted therapies and chemotherapies, attempt to overcome immunosuppression in tumors. The application of immune checkpoint inhibitors provides a novel tool for the development of combined therapy approaches [14].

### 2.3. Poor Quality of the Tumor Vasculature

Tumorigenesis starts with the formation and uncontrolled growth of transformed cells. In developing tumors, the supply with dioxygen and nutrients is diminished due to the poor quality of the tumor vasculature, longer diffusion paths, insufficient lymphatic drainage, fluctuations in interstitial pressure, and intermittent vascular collapse in contrast to healthy tissues [68,69].

In solid tumors, the growth factor VEGF is overexpressed, whereby hypoxia promotes this expression [70,71]. This growth factor is responsible for the poor quality of the tumor vasculature with irregular, leaky, and immature vessels [72,73]. In addition, the VEGF is a key angiogenic factor in tumors and is involved in tumor progression and metastasis [73,74].

### 2.4. Direct Effects of Hypoxia in Tumor Cells

In tumors, hypoxia [75,76,77] provokes significant metabolic alterations in tumor cells, such as the activation and stabilization of hypoxia-inducible factor 1α (HIF-1α) [78], upregulation of glycolysis [79], downregulation of oxidative phosphorylation in the mitochondria [79], and enhanced formation of reactive species by dysfunctional mitochondria [80]. In addition, lactate accumulates [81,82] and induces numerous lactate-driven effects [81,83,84].

Cytosolic HIF-1α is a master regulator of glycolysis in many cells. This usually short-lived factor is controlled by prolyl hydroxylase that tags HIF-1α for proteasome degradation [85,86]. Hypoxia, stress-induced oxidation of Fe^2+^ in propyl hydroxylase, and an ascorbate deficiency prevent HIF-1α degradation [78]. As a result, HIF-1α upregulates several enzymes promoting glycolysis and downregulates pyruvate dehydrogenase that supplies acetyl-coenzyme A (Ac-CoA) for the citrate cycle [79]. Unlike HIF-1α, which is a ubiquitous protein, HIF-2α is predominantly expressed in highly vascularized tissues [87]. HIF-2α is active during prolonged hypoxia, replaces HIF-1α in a spatiotemporal manner [75,88], and is involved in controlling oxidative stress, the cell cycle, blood vessel remodeling, and RNA transport [89].

In healthy cells, lipogenesis is mostly induced by pyruvate-derived Ac-CoA. To ensure lipogenesis in hypoxic cancer cells, decreased pyruvate is replaced in the TCA cycle by glutamine, supporting the conversion of α-ketoglutarate into citrate and Ac-CoA [90]. In cancer cell growth, glutamine is an important carbon and nitrogen source for lipid, amino acid, and nucleotide synthesis [91].

In tumorigenesis, increased lactate production by tumors cells reprograms macrophages, T cells, and other immune cells in a way that they are immunosuppressive and anti-inflammatory [81,82,83,84]. Lactate also promotes angiogenesis and tumor progression [83,92], supporting the hyaluronan release from the adjacent fibroblasts. This hyaluronan cover protects additional tumor tissue from immune attacks [93].

### 2.5. Stress-Related Responses in Tumor Cells

In tumor cells, enhanced intracellular levels of reactive species, which result mainly from dysfunctional mitochondria, activate the transcription factor nuclear factor-erythroid 2-related factor 2 (Nrf2) that is low expressed under normal physiological conditions [94,95]. In cancer cells, Nrf2 promotes the syntheses of numerous enzymes involved in antioxidative defense and contributes to a resistance against chemo- and radiotherapy (see Section 3.3).

Prolonged hypoxia also mediates the processes of autophagy [96,97]. In the early stages of tumorigenesis, autophagy is directed to inhibit tumor growth [98,99,100]. Otherwise, in late-stage cancers, autophagy is known to stabilize cancer cells by maintaining the integrity of the mitochondria, reducing DNA damage, and increasing the resistance against stress. Under hypoxic conditions and a low supply of nutrients, autophagy can provide energy resources for the survival of cancer cells and resistance against chemotherapy [101]. With these mechanisms, autophagy contributes to tumor development [102,103,104] and facilitates metastasis [105,106,107,108].

Under stress situations, the percentage of misfolded proteins increases in the endoplasmic reticulum (ER) as a result of the action of reactive species and electrophiles. Misfolded proteins are subjected to ubiquitinylation with subsequent proteasomal degradation or autophagy. Three major stress sensors of an unfolded protein response (UPR) are activated by misfolded proteins in a wide range of tumor cells [109]. The overexpression of chaperones or mutations in the UPR pathways are used by tumor cells to antagonize ER stress [110,111,112,113]. A hyperactive UPR promotes tumorigenesis in advanced cancers [114].

To maintain a reducing environment in the cytosol of tumor cells, the enhanced uptake of cystine/cysteine by tumors and the conversion into glutathione are mandatory to protect cells from stress-mediated cell death [115]. In tumors, highly increased extracellular cysteine levels are observed, whereas cysteine is slightly enhanced within tumor cells [116]. Moreover, cysteine is mainly transported into tumor cells by the cystine/glutamate antiporter solute carrier family 7 member 11 (SLC7A11), which is widely expressed in human cancers [117,118]. In tumor cells, the conversion of cysteine into hydrogen sulfide activates the Nrf2-mediated gene expression of antioxidative proteins [119,120]. Hydrogen sulfide also accelerates the cell cycle in tumor cells [121].

In addition, a close link exists between the upregulation of reductive glutamine metabolism, also known as reductive carboxylation, and the transport of NADPH from cytosol into the mitochondria to upregulate mitochondrial GSH and protect against reactive species [122]. These pathways are important for anchorage-independent growth in cancer cells.

### 2.6. Intratumoral Hemorrhages

The rupture of blood vessels results in intratumoral hemorrhages, which are typical of many tumors [123,124]. The conditions in hemorrhages promote the formation of cytotoxic free heme (see Section 4 for more details) and in turn the transcriptional processes in tumor cells via the binding of free heme to guanine-rich DNA and RNA structures (known as G4 elements) [125,126]. Free heme controls the expression of key target genes, including telomeres and oncogenes (such as *c-Myc*), and favors malignant transformation [127,128].

### 2.7. Reversed pH Gradient in Cancer Cells

In cancer cells, the cytosolic pH is enhanced to 7.3–7.6 (versus about 7.2 in normal cells), whereas the extracellular pH is decreased to 6.8–7.0 (versus about 7.4 in normal cells) [129,130]. In the TME, the milieu is more acidic by about 0.3–0.7 pH units in comparison to the cytosolic pH of tumors cells [131]. Thus, the pH gradient across the plasma membrane of cancer cells is reversed in contrast to healthy cells. These deviations are mediated by hypoxia and lactate effects and accompanied by an increased expression and activity of ion transporters in the plasma membrane and intracellular pH regulators [132].

In many cell types, enhanced intracellular pH values by ~0.3–0.4 units are found in proliferating cells at the end of the S phase [133,134] and in migrating cells [135,136]. The reverse pH gradient in cancer cells promotes cell proliferation, survival, migration, and metastasis [129,130,137,138].

### 2.8. Matrix Remodeling in the TME

In the TME, the extracellular matrix (ECM) acts as a barrier around tumor cells against cytotoxic immune cells. Under hypoxic conditions, the stiffness of the matrix components, especially of collagen fibrils, increases due to more intense cross-linking reactions between matrix polymers [139,140]. These alterations are mainly caused by cancer-associated fibroblasts [141]. Several enzymes involved in matrix remodeling and cross-linking reactions are upregulated in tumors, namely lysyl oxidase, lysyl oxidase-like proteins, collagen prolyl 4-hydroxylase, and WINT1-inducible signaling pathway protein 1 (WISP1) [142,143,144,145].

Tumor fibrosis is closely associated with an overexpression of these enzymes [146]. As a result, ECM fibers, most of all collagens I and III, are converted into dense, linearized, and cross-linked fiber bundles with altered mechanical properties [147,148,149,150]. The enhanced stiffness of the ECM components favors the migration of single tumor cells, promotes angiogenesis, and dampens anti-tumor activities [151]. The elevated degree of tumor fibrosis is associated with a poor prognosis in many types of cancer [151,152,153].

In the TME, the deposition of the matrix components is important for angiogenesis, proliferation, tumor cell invasion, and metastasis. Several matrix metalloproteases such as MMP-2, -3, -9, and -14 are upregulated in malignant tumors [154,155]. An enhanced expression of heparinase also favors angiogenesis [156,157,158].

The enhanced proliferation of tumor cells is also supported by increased hyaluronan production [159,160,161]. Under hypoxic conditions, the production of hyaluronan is markedly increased by tumor cells [162]. M2-type macrophages accumulate preferentially in hyaluronan-rich areas of the TME [163]. In addition, hyaluronan favors the phenotype change in monocytes and macrophages to the M2-type [164].

## 3. Cytotoxic Agents in Cancer Cells

### 3.1. Sources for Cytotoxic Agents in Cancer Cells

The shift in energy metabolism toward the dominance of glycolysis in tumor cells markedly affects the properties of the mitochondria. The tumor-associated dysfunction of the mitochondria has multiple facets, including mutations of mitochondrial DNA, defective enzymes in the TCA cycle, glutamine-induced reductive carboxylation in the TCA cycle, disturbed electron redox conversions, imbalances in the signaling pathways, resistance to apoptosis induction, and disturbed energy metabolism [165]. Dysfunctional mitochondria are a major source for reactive species, such as the superoxide anion radical (O_2_^•−^) and hydrogen peroxide (H_2_O_2_) [80]. Highly reactive species can be derived from the reactions of both these species.

In tumor cells, several mechanisms contribute to the enhanced values of catalytically active ferrous ions (Fe^2+^). Iron overload is involved in cell death by ferroptosis [166,167] and in the formation of highly reactive hydroxyl radicals (HO^•^).

Another source of cytotoxic agents in tumors is free heme that results either from degradative processes in intratumoral hemorrhages or from defective cellular heme proteins. Apart from numerous cytotoxic activities, free heme promotes the oncogene activation via interactions with heme-responsive elements in promoter regions [127,128].

O_2_^•−^ and H_2_O_2_ are less reactive species that can react with metal ions, radical species like NO, or iron-containing proteins to generate very reactive and tissue-damaging agents such as hydroxyl radical (HO^•^), and peroxynitrite (ONOO^−^) [168]. Although many (patho)physiological processes are associated with the excess formation of reactive species in living tissues, in tumor cells the main focus is directed on dysfunctional mitochondria. Other physiologically relevant highly reactive species are hypochlorous acid (HOCl), hypobromous acid (HOBr), and singlet oxygen (^1^O_2_). HOCl and HOBr are generated in H_2_O_2_-driven reactions by heme peroxidases from activated neutrophils and eosinophils [169,170,171]. The formation of ^1^O_2_ is relevant to surface-exposed locations like the skin and eyes, where ultraviolet and visible light can induce photooxidative processes [172,173].

### 3.2. Mitochondria-Derived Reactive Species in Cancer Cells

In intact mitochondria, only a small percentage of O_2_ is converted into O_2_^•−^. The main mitochondrial sources of O_2_^•−^ are redox conversions in complex I and the ubiquinone cycle [174]. Spontaneous and superoxide dismutase (SOD)-catalyzed dismutation of two O_2_^•−^ yields H_2_O_2_ and O_2_. In dysfunctional mitochondria, an enhanced production of O_2_^•−^ and H_2_O_2_ is favored by reverse electron transport in complex I [175] and by disturbances in the electron flow in the ubiquinone cycle [176].

In the mitochondria, apart from dismutation reactions, O_2_^•−^ is involved in two major pathways, both contributing to the formation of highly reactive species, which exhibit cytotoxic activities. The reaction of O_2_^•−^ with mitochondrial aconitase and other proteins containing [4Fe-4S]^2+^ clusters is associated with the release of catalytically active Fe^2+^ [177,178], which is involved in the formation of HO^•^.

The second pathway concerns the reaction between O_2_^•−^ and nitric oxide (NO), which is synthesized by NO synthases. The mitochondria exhibit NO synthase activity [179]. However, which isoform is present in the mitochondria is still unsolved [180,181]. Otherwise, NO can also be derived from other cellular sources, as this agent diffuses well through membranes. Inducible NO synthase (iNOS) is upregulated in multiple cancers such as breast, colorectal, prostate, esophageal, and gastric cancers [182,183,184]. The very rapid reaction between NO and O_2_^•−^ promotes the formation of ONOO^−^ [185,186]. NO competes with SODs for O_2_^•−^ and lowers the load of O_2_^•−^ within the mitochondria [187,188]. ONOO^−^ is involved in the nitration of tyrosine residues, the formation of thiyl radicals and radical species after reaction with CO_2_, and initiates the lipid peroxidation processes [189,190,191,192]. ONOO^−^ is scavenged and inactivated by glutathione peroxidases (GPXs), especially GPX1, heme proteins, and heme peroxidases [193,194,195,196]. Both NO and ONOO^−^ inhibit several enzymes of the mitochondrial respiratory chain [179].

H_2_O_2_, which mainly results from the dismutation of O_2_^•−^, is known to react with free transition metal ions, such as Fe^2+^ and Cu^+^, to yield the highly reactive HO^•^ in the so-called Fenton reaction [197,198,199]. HO^•^ reacts as nearly diffusion controlled with many neighbored biological substrates. Thus, no enzymatic mechanism exists to scavenge and inactivate this radical [200]. Therefore, the main physiological strategy against HO^•^ is to avoid its formation through the proper control of iron and copper metabolism during the transport, storage, and utilization of free metal ions [201,202,203,204,205,206,207].

The major antagonizing principles against small reactive species are summarized in reference [16]. O_2_^•−^ is deactivated by SODs and cytochrome c [208,209,210,211,212,213]. Whereas SOD1 is found within cells at three distinct locations: cytosol, intermembrane space of mitochondria, and nucleus [210]. Within the intermembrane space of the mitochondria, O_2_^•−^ is also detoxified by cytochrome c [212,213]. H_2_O_2_ is controlled by GPXs, peroxiredoxins, and catalase [214,215,216,217]. Unlike O_2_^•−^, H_2_O_2_ is able to permeate through biological membranes. Thus, the defense reactions against H_2_O_2_ concern the whole cell. These mechanisms hold down their concentrations in biological media and prevent dangerous side reactions of these species.

### 3.3. Antioxidant Defense Mechanisms and Stabilization of Redox Homeostasis in Cancer Cells

The enhanced content of reactive species in malignant cells, which mainly results from dysfunctional mitochondria, is associated with oxidative stress and the upregulation of antioxidative mechanisms [80]. Hypoxia stimulates the production of reactive species in the mitochondria. In turn, H_2_O_2_ stabilizes HIF-1α [218,219,220]. The participation of reactive species in carcinogenesis is supported by several studies [221,222,223,224,225]. These species affect anchorage-independent cell growth, epithelial-to-mesenchymal transition, angiogenesis, and apoptosis [226]. As high levels of reactive species are lethal, numerous antioxidative mechanisms are upregulated in cancer cells [227,228].

A key sensor for oxidative stress is Nrf2 that is low expressed under normal physiological conditions. Cytosolic Nrf2 is under the control of a Kelch-like erythroid cell-derived protein, with CNC homology [ECH]-associated protein 1 (Keap1) and Cul3 forming a complex with Nrf2 and inducing its ubiquitination and proteasomal degradation [229]. Under oxidative and electrophilic stress, Nrf2 is released from this complex and can mediate transcriptional processes in the nucleus. Nrf2 promotes the synthesis of several enzymes involved in the detoxification of reactive species (thioredoxin 1, thioredoxin reductase 1, thioredoxin interacting protein, peroxiredoxins, glutaredoxins, glutathione peroxidase, glutathione reductase, sulfiredoxin), the biosynthesis of glutathione, enzymes involved in the phase II and phase III inactivation of xenobiotics, and proteins controlling iron metabolism [94,95]. Enzymes such as glucose-6-phosphate dehydrogenase and 6-phosphogluconate dehydrogenase involved in the pentose phosphate pathway for the generation of NADPH are also upregulated by Nrf2 [230]. This pathway also produces ribulose-5-phosphate, which is necessary for enhanced nucleotide synthesis in malignant tumors [231].

Nrf2 is able to inhibit the initiation of carcinogenesis. Otherwise, it promotes the progression of already transformed cells [232,233]. In several human cancers, Nrf2 is stabilized by a reduced Nrf2-Keap1 interaction due to missense mutations in the *keap1* and *Nrf2* genes [228,234,235,236]. In cancer cells, Nrf2 can be activated by mutations or the epigenetic silencing of Keap1 as well as by the transcriptional activation via oncogenes like *k-ras*, *c-myc*, or *b-raf* [237].

Redox homeostasis is based on the production of NADPH as a universal source of electrons [238,239]. As a cofactor of glutathione reductase and thioredoxin reductase, NADPH ensures the recovery of GSH from oxidized glutathione (GS-SG) and reduced thioredoxin (Trx(-SH, -SH)) from the oxidized form (Trx(-S-S-)) [240]. GSH is an essential element in GPX-mediated reactions, the reduction in disulfides, and the reduction in glutaredoxin. Thioredoxin is involved in the reduction in peroxiredoxins, in ribonucleotide reductase-mediated reactions, and the recovery of methionine sulfoxides by methionine sulfoxide reductases (MSRs). With these reactions, NADPH ensures the maintenance of the thiol and methionine groups in cellular proteins and small molecules like GSH [238,241].

At the initiation of tumorigenesis, antioxidative systems may exert an anti-tumor role, as shown in the examples for peroxiredoxins [226], components of the thioredoxin system [242], and glutathione peroxidase 1 [243]. However, in growing tumors, the antioxidative mechanisms are essential for protecting tumor cells from excessive damage and apoptosis induction by reactive species. In late-stage cancer cells, a higher expression of NrF2, HIF-1α, and UPR promotes a marked upregulation of the aforementioned intracellular antioxidative defense mechanisms and systems promoting the maintenance of redox homeostasis, which makes cancer cells insensitive against oxidative stress and additional stress induced by chemo- and radiotherapy. Examples of the upregulation of these mechanisms in late-stage tumors are listed in Table 1. An overview of the major protective systems in cancer cells is also given in Figure 3.

Few remarks are necessary concerning the detoxification of methionine sulfoxides, which exists in two chiral forms, the *R*- and *S*-form of methionine sulfoxide. MSR-A is specific to the *S*-form of methionine sulfoxides. The downregulation of MSR-A is associated with a more aggressive phenotype of cancer cells in breast cancer [244]. In contrast, the upregulation of the MSR-B types, which reduce the *R*-forms of methionine sulfoxides, contributes to cancer growth in different cancer types [245,246,247].

As the upregulation of the aforementioned antioxidative mechanisms contributes to tumor growth and disease progression, the pharmacological inhibition of these enzymes is applied as a potential therapeutic trial against cancers.

**Table 1 cancers-16-00081-t001:** Overexpression of the defense systems against reactive species and systems maintaining redox homeostasis in cancer cells of late-stage cancers.

Antioxidative System	Functions	Cancer Types	References
SOD1	Removal of O_2_^•−^, formation of H_2_O_2_	Lung and breast cancer	[248,249]
SOD2	Removal of O_2_^•−^, formation of H_2_O_2_	Esophageal cancer and others	[250,251,252]
Catalase	Removal of H_2_O_2_	Glioblastoma	[253]
Peroxiredoxins (six subtypes)	Removal of H_2_O_2_	Many types of tumors	[226,254]
Thioredoxin system (thioredoxin, thioredoxin reductase; NADPH)	Removal of H_2_O_2_, redox homeostasis	Lung, breast, prostate, colorectal, pancreatic, hepatocellular, gastric, and other carcinomas	[255,256]
Cystine/glutamate anti-porter solute carrier family 7 member 11	Cystine/cysteine uptake in cells for GSH production	Multiple types of cancer	[117,118]
Glutathione (GSH)	Redox homeostasis, cofactor of glutathione peroxidases	Ovarian, breast, and lung cancers	[257,258,259]
Glutathione reductase	Recovery of GSH	Glioblastoma	[260]
Glutathione peroxidase 4	Removal of H_2_O_2_ and lipid hydroperoxides, inhibition of ferroptosis	Thyroid, colorectal, and esophageal carcinomas	[261,262]
Glutathione peroxidase 1	Removal of H_2_O_2_, peroxynitrite, and lipid hydroperoxides	Breast, hepatocellular, colorectal, esophageal, and lung carcinomas	[243]
Glutathione peroxidase 2	Removal of H_2_O_2_	Liver, colorectal, breast, lung, and prostate carcinomas	[263,264]
Glutaredoxins 3 and 5	Redox homeostasis	Liver and colorectal cancers	[265]
sulfiredoxin	Redox homeostasis	Skin, liver, and colorectal cancers	[266]
Ribonucleotide reductase	Redox homeostasis	Gastric, ovarian, bladder, colorectal cancers	[267,268]
Methionine sulfoxide reductase B1 and B3	Redox homeostasis	Hepatocellular carcinoma and others	[245,246,247]
Glucose-6-phosphate dehydrogenase	NADPH production, redox homeostasis	Multiple types of cancer	[269,270]
6-phosphogluconate dehydrogenase	NADPH production, redox homeostasis	Multiple types of cancer	[271,272]

### 3.4. Disturbed Iron Homeostasis in Cancer Cells

As redox-sensitive iron ions (Fe^2+^ and Fe^3+^) are involved in tissue-damaging reactions like Fenton reaction, all the aspects of iron metabolism are strictly controlled to avoid an increase in free iron ions [273,274]. In blood, iron ions are mainly sequestered by transferrin, which is the ligand to the transferrin receptor on the cell surfaces [202,203]. In cells, iron is stored inside ferritin, which represents a hollow sphere [204].

Malignant cancer cells require iron for all the processes of tumor progression. Transferrin receptor 1 is highly upregulated in cancer cells [275]. Ferritin is overexpressed in esophageal adenocarcinoma, glioblastoma, and breast and colorectal carcinomas [276]. The iron exporter protein ferroportin has a lower expression in prostate and breast cancers and is associated with a poor prognosis in pancreatic cancer [277,278,279]. These mechanisms contribute to a higher iron load in cancer cells and concomitant protection against iron-mediated damaging reactions. These data are summarized in Table 2 and included in Figure 3.

### 3.5. Intratumoral Hemorrhages and Free Heme

Intratumoral hemorrhages are a common clinical manifestation in patients with lung, bladder, gastric, or colorectal cancers [124]. They result from a rupture of leaky blood vessels. In these hemorrhages, hemoglobin is released from defective red blood cells and rapidly oxidized to methemoglobin, which is unable to bind dioxygen. This oxidation is mostly favored by NO. Thus, an excessive release of hemoglobin diminishes the bioavailability of NO [283,284]. In circulating blood, methemoglobin is scavenged by haptoglobin and the resulting haptoglobin–heme protein complex is removed by binding to CD163 expressed on liver and spleen macrophages [285,286].

In a similar way, myoglobin released from damaged muscles is oxidized to metmyoglobin, which is eliminated from blood via scavenging by haptoglobin [287,288]. Both methemoglobin and metmyoglobin are known to liberate ferric protoporphyrin IX, also known as free heme or labile heme [289]. In blood, free heme is antagonized by hemopexin through the formation of a high-affinity complex, which is cleared by hepatocytes via binding to CD91 [290].

In the context of inflammatory processes, free heme acts both as a DAMP being a ligand to TLR4 [291,292] and as a strong cytotoxic agent [16]. It promotes numerous cytotoxic effects, such as the induction of oxidative processes in membranes, lipoproteins, and hydrophobic areas of proteins [293,294], the induction of hemolytic processes in unperturbed red blood cells [295,296], and promotes serious damage in the kidneys and liver [297,298]. Serious health problems are associated with excessive intravascular hemolysis and rhabdomyolysis leading to a limited availability of haptoglobin and especially hemopexin. Elevated levels of free heme are found in malaria, sickle cell disease, and porphyria [299].

In tumors, hemorrhages and enhanced hemoglobin levels are associated with a poor clinical outcome for affected patients [124,300,301]. An inverse correlation between the level of free heme and hemopexin was found in the blood of patients with prostate cancer [299]. As shown in prostate cancer cells, external free heme is taken up by these cells and accumulates in the nucleus, where it intercalates into parallel guanine-rich quadruplex DNA and RNA structural elements (so-called G4 elements) [125,126]. In this way, free heme promotes the expression of heme-responsive genes that are involved in heme degradation and iron sequestration. Intercalating heme also controls the expression of the key target genes, including telomeres and oncogene promoters [127,128]. Free heme targets the promoter region of the potent oncogene *c-Myc* [299] that is expressed in about 80% of human malignancies [302].

In hemopexin knockout mice, the most aggressive tumor phenotype was detected supporting the view that free heme promotes tumor growth and metastasis in prostate cancer [299]. Free heme also induces the accumulation of nuclear heme oxygenase-1, which contributes to the unwinding of the G4 elements and an increase in the genomic instability in cancer cells [303].

### 3.6. Protection against Free Heme by Heme Oxygenase 1 (HO-1)

Within cells, free heme is detoxified by HO-1 in an O_2_ and NADPH-dependent reaction to yield carbon monoxide (CO), Fe^2+^, and biliverdin, which is further metabolized by biliverdin reductase to bilirubin [304,305].

In tumor cells, HO-1 exhibits anti-oxidative, anti-inflammatory, anti-apoptotic, and pro-angiogenic effects [306]. HO-1 also activates autophagy and contributes to the chemoresistance of breast cancer cells [307]. During tumorigenesis, HO-1 is highly upregulated in multiple cancer types, including breast, prostate, liver, gastric, pancreatic, colorectal, and esophageal cancers, and neuroblastoma (Table 2) [282]. This upregulation is associated with therapy resistance and a poor prognosis for patients. Within the mitochondria, HO-1 protects these organelles from mitochondrial free heme and prevents apoptotic cell death [308].

## 4. Cytotoxic Agents in the TME

### 4.1. Special Conditions in the TME

Tumor cells are encompassed by the TME, which contains numerous cell types, including mesenchymal stem cells, fibroblasts, endothelial cells, and a variety of immune cells as well as extracellular matrix (ECM) components. Unlike healthy tissue, the conditions in the TME are dictated by hypoxia, an enhanced basic level of H_2_O_2_, acidosis, deficiencies in essential nutrients and metabolites, and by the presence of bioactive agents such as inflammation-resolving cytokines, growth factors, and mediators released from tumor and stromal cells. Tumor cells affect their environment through the release of exosomes containing numerous active agents that promote tumor progression, immunosuppression, and remodeling of the TME. Exosomes are also involved in the angiogenesis, migration, and invasion of cancer cells, and play a role in the formation of a metastatic niche [41,309,310]. Altogether, in advanced cancers, special conditions in the TME contribute to tumor growth and progression.

The presence of TGF-β, which is derived from tumor cells and some other cells in the TME, is a key factor affecting many aspects of tumorigenesis. Although TGF-β acts as a tumor suppressor in the early stages of disease, it develops to a potent tumor promoter in the late stages of cancer [311,312,313,314]. In the TME, TGF-β is a major metabolic driver and plays a crucial role in the reprogramming of immune cells favoring immune tolerance and cancer progression [313,315,316]. Moreover, TGF-β is essentially involved in the development of resistance against anti-cancer therapies. Thus, the inhibition of TGF-β signaling in combination with other therapeutic approaches should be a promising tool for the treatment of therapy-resistant cancer forms [313].

### 4.2. Peculiarities of Cytotoxic Agents in the TME

In the TME, the contribution of cytotoxic agents and their protective mechanisms in tumorigenesis is more complex than in tumor cells. In addition to oxidant-based cytotoxic agents like reactive species and free heme, proteases like elastase, matrix metalloproteases (MMPs) also exhibit enhanced activities in comparison to healthy tissue areas. This also concerns angiotensin II, which is generated from angiotensin I and angiotensinogen by serine proteases. In the TME, cytotoxic agents are derived from stromal cells and immune cells, especially from neutrophils and G-MDSCs. Neutrophils and neutrophil subpopulations are able to generate O_2_^•−^ and H_2_O_2_ and activate the heme protein myeloperoxidase, which exhibits a pronounced halogenation and peroxidase activity [169,317,318]. These cells also secrete the serine proteases elastase, cathepsin G, and proteinase 3 [319]. Specifically, elastase has a wide range of substrates, including the components of the ECM.

However, the status of protective mechanisms against cytotoxic agents is not well defined. In the TME, other protective principles dominate against cytotoxic agents in contrast to tumor cells. The dominating protective systems are the proteins controlling the transition free metal ions (ceruloplasmin, transferrin), proteins avoiding the formation of free heme (haptoglobin, hemopexin), antiproteases inhibiting serine proteases, TIMPs inactivating MMPs, and small antioxidative acting molecules (urate, ascorbate, polyphenols). Some of the mentioned proteins are acute-phase proteins. As the data on these antagonizing principles are usually determined in the serum and other fluids of cancer patients, these values only badly reflect real conditions in pericellular areas of tumor cells. Furthermore, the TME does not represent a unique structure. Instead, it is a poor perfused region with multiple gradients for dioxygen, nutrients, and metabolites towards the tumor center.

Often protective mechanisms are more expressed in cancer patients in comparison to healthy individuals as found, for example, in ceruloplasmin [320,321], haptoglobin [322], hemopexin [323], and some protease inhibitors [324,325,326]. This higher expression can be explained as a response to the ongoing inflammatory process. However, in some cases, opposed data are obtained for some protective mechanisms with a dependence on certain cancer types. The selected examples are diminished haptoglobin values in hepatocellular carcinoma [327], reduced hemopexin levels in prostate cancer patients [299], and downregulated ceruloplasmin in adrenocortical and hepatocellular carcinomas [328,329]. In addition, the expression of protective systems depends not only on the presence of tumors but is also related to a patient’s specific physiological status and existing non-tumorous diseases. Thus, it remains very challenging to draw meaningful conclusions for the balance between TME-related cytotoxic agents and antagonizing principles in cancer patients.

By analyzing the haptoglobin protein expression in patients with hepatocellular carcinoma using immunohistochemical staining, it was observed that this protein was highly expressed in adjacent non-tumorous cells, but rarely detectable in the tumor tissue [327]. Within the tumor, a very low haptoglobin level was found in tumors with poorly differentiated cancer, a condition with poor prognosis [327]. This study showed that great differences may exist for the expression of protective principles between tumors and healthy tissue regions.

In addition to the upregulation or downregulation of the protective principles, strategies for tumors against cytotoxic agents in the TME also comprise the depletion of essential metabolites and nutrients as well as an enhanced production of the ECM components.

### 4.3. Neutrophil Subpopulations with the Tumor-Promoting Phenotype

Mature neutrophils are known for their anti-tumoral activities [330,331,332]. In contrast, neutrophilia and a high ratio between neutrophils and lymphocytes are associated with a poor clinical outcome in many cancers [333].

In circulating blood of cancer patients, two subpopulations of neutrophils accumulate, which both exhibit pro-tumoral characteristics. Apart from the aforementioned G-MDSCs, which represent an immature phenotype of granulocytes, mature neutrophils are converted under the action of TGF-β released from tumor cells and different stromal cells into a tumor-promoting phenotype [334]. The latter type has been called N2-type neutrophils in analogy to macrophage phenotypes. Both G-MDSCs and N2-type neutrophils have a lower density than mature neutrophils.

In the expansion and activation of MDSCs, the activation of STAT3 plays a critical role. This transcription factor mediates the expression of the arginase 1, IL-10, PL-D1, S-100 proteins, which are all involved in immunosuppressive activities [32,35]. Moreover, the STAT3 expression is regulated by several microRNAs [335,336,337].

### 4.4. Involvement of Myeloperoxidase (MPO) in the Immunosuppression by G-MDSCs

In cancers, the production of reactive species by G-MDSCs mediates the immunosuppressive activity against CD8^+^ T cells [338,339]. Murine G-MDSCs exhibit a higher activity of arginase, MPO, and intracellular reactive species than murine neutrophils [339]. The MPO activity is lower in MDSCs from tumor-bearing mice compared to MDSCs from tumor-free mice [338].

The immunosuppressive function of G-MDSCs is related to the transfer of lipid bodies with oxidatively truncated lipids to DCs, where these lipids inhibit the cross-presentation of tumor-associated antigens and impair anti-tumor CD8^+^ T cell responses unlike lipid bodies with non-oxidized lipids [340]. Thereby, the interaction of modified lipids with chaperones blocks the antigen cross-presentation in DCs [341]. In G-MDSCs, a concerted action of NADPH oxidase and MPO is responsible for the formation of oxidized, truncated lipids in G-MDSCs. The involvement of MPO was evidenced in the experiments using MPO-knockout mice and a pharmacological inhibition of MPO [340]. It remains puzzling how MPO favors the oxidation of lipids in G-MDSCs. The immunosuppressive activities of G-MDSCs depend on the uptake of lipids via the scavenger receptor CD36 and with the help of the fatty acid transport protein 2, as fatty acid oxidation is the main energy source in tumor-associated G-MDSCs [342,343,344]. Apparently, MPO oxidizes lipid droplets inside G-MDSCs.

In another study, both genetic and pharmacological blockades of MPO increased the success of immune checkpoint therapy in experimental models with primary melanoma [345]. There was a significant increase in MPO in melanoma tissue arrays. However, no details were given regarding how MPO contributed to the resistance against immune checkpoint therapy [345].

### 4.5. Elastase-Mediated Effects in the TME

In a serum-free incubation medium, elastase released from human neutrophils is able to kill a large variety of cancer cells [346]. However, these data are irrelevant as the role of important elastase substrates, like serum antiproteases and components of the extracellular matrix, were not considered in these experiments. The current view is that elastase promotes tumorigenesis and metastasis [347]. This was evidenced by applying elastase knockout mice and a pharmacological inhibition of elastase in cancer models [347,348,349]. A deficiency of α_1_-antitrypsin, an efficient antagonist against elastase, is a risk factor for the development of hepatic and lung cancer [350,351].

Moreover, elastase is significantly enhanced in several tumors [352,353,354]. This protease is mainly derived from invading neutrophils and myeloid-derived suppressor cells. The detrimental activity of elastase in tumors is associated with a proteolytic degradation of the components of the vessel wall and extracellular matrix. This facilitates tumor cell intravasation and dissemination via elastase-mediated dilated intratumoral blood vessels [347]. Elastase has a diversified substrate spectrum, including elastin, laminin, and transmembrane proteins (E-cadherin, VCAM-1, G-CSF receptor), cytokines (IL-1, G-CSF), vascular endothelial growth factors (VEGFs), and precursors of angiotensin II [347]. It activates MMP2, MMP3, and MMP9 from inactive precursors, and degrades TIMP-1 contributing to ECM degradation [355,356,357]. In the TME, some potential elastase substrates are overexpressed, like VEGFs [70,71], several MMPs [154,155], elastin [358,359], and laminins [360].

The inhibitory activity of antiproteases depends on two factors. First, the oxidation of critical residues at active site of antiproteases is associated with a loss to inhibit elastase. In α_1_-antitrypsin, these critical residues are Met-351 and Met-358 [361]. Second, the cationic elastase is known to bind to negatively charged components of the extracellular matrix and surfaces. Matrix- and surface-associated elastase cannot be inhibited by antiproteases [362]. In tumors, the high abundance of matrix components apparently favors the enhanced activity of elastase.

In the same cancers, the overexpression of antiproteases is associated with tumor progression and metastasis, as found for elafin in hepatocellular carcinoma [325], the secretory leukocyte protease inhibitor in colorectal cancer [324], or α1-chymotrypsin in pancreatic cancer [326]. Otherwise, the enhanced α1-chymotrypsin expression promotes survival in liver cancer [326].

### 4.6. Role of Angiotensin II

As part of the renin-angiotensin-aldosteron system, angiotensin II contributes to the regulation of blood pressure and water metabolism. It is formed under the action of the angiotensin converting enzyme (ACE) from the decapeptide angiotensin I. Under inflammatory conditions, neutrophil-derived serine proteases cathepsin G, elastase, and proteinase 3 release angiotensin II from angiotensin I and angiotensinogen [363,364,365]. Angiotensin II promotes actin cleavage, proteolysis, increased apoptosis, caspase-3 activation, and the lowering of IGF-1 [366,367]. Further effects of angiotensin II are the activation of NADPH oxidase, the production of reactive species, such as O_2_^•−^ and H_2_O_2_ [366,367], and the promotion of the ubiquitin-proteasome proteolytic pathway [368].

Angiotensin II is under the control of ACE2, which converts it into anti-inflammatory angiotensin 1–7. The inhibition of ACE2 by coronaviruses, including SARS-CoV-2, enhanced the impact of angiotensin II on lung damage [369,370].

The effects of angiotensin II are mediated by ligation to angiotensin II receptors (ATR), most of all to ATR1. The overexpression of this receptor plays a significant role in cancer aggressiveness and angiogenesis in various kinds of cancers, such as ovary, bladder, lung, and breast cancers [371,372,373,374,375,376].

Angiotensin receptor blockers are widely used as medication for the treatment of hypertension, heart failure, diabetic nephropathy, and to prevent cardiovascular events [377,378]. The risk of cancer, especially of lung cancer, increases with treatment using these drugs [379].

### 4.7. NO Metabolism in the TME

Generally, the NO effects on tumors are concentration dependent. At low concentrations, NO promotes tumor progression via apoptosis inhibition and the enhancement of angiogenesis and metastasis through the activation of several survival pathways. At higher concentrations, NO becomes cytotoxic via nitrosative stress, DNA damage, and apoptosis induction [380,381].

Several factors reduce the bioavailability of NO in tumors (Figure 4). Arginase 1 and arginase 2 compete with NO synthases for L-arginine, limiting the production of NO. Both arginases are overexpressed in various cancer types, like breast, gastric, colorectal, and liver cancers [382,383,384,385]. Arginases produce L-ornithine, which is further metabolized to yield proline for collagen synthesis and polyamines involved in tumor cell proliferation, migration, and invasion [386]. In the TME, arginases from tumor cells, MDSCs, macrophages, and Tregs contribute to the depletion of L-arginine and impair T cell functions and tumor perfusion [387].

In tumor hemorrhages, the release of hemoglobin from leaky red blood cells also contributes to a reduced bioavailability of NO as the oxidation of hemoglobin to methemoglobin is usually promoted by NO [283,284]. Methemoglobin, which is under the control of haptoglobin, liberates cytotoxic free heme. When not antagonized by hemopexin, this agent is able to promote the release of hemoglobin from yet unperturbed red blood cells [289,290], diminishing the bioavailability of NO.

Further, NO reacts very rapidly with O_2_^•−^ to yield peroxynitrite [185,186]. In addition to the production of O_2_^•−^ by dysfunctional mitochondria inside tumor cells, O_2_^•−^ can also result from the activity of NADPH oxidase (also known as NOX2) present in neutrophils and many other cells in the TME [388,389], from xanthine oxidase-mediated reactions [390,391], and from the autoxidation of hemoglobin and myoglobin [392,393]. In the extracellular medium, O_2_^•−^ is under the control of SOD3, the secretory form of SODs. Although SOD3 is found in many tissues, higher SOD3 values are known for their well vascularized tissues, such as lung, kidney, uterus, placenta, and blood vessels [394]. In many tumors, SOD3 is downregulated and promotes cancer metastasis and malignant progression [395,396,397,398]. A reduced SOD3 expression diminishes the bioavailability of NO, as NO reacts easily with O_2_^•−^ to yield peroxynitrite. The latter agent damages the components of the ECM and endothelial cells, contributes to leaky blood vessels in tumors, and reduces the invasion of T cells into tumors [399,400]. Peroxynitrite is also able to deactivate antiproteases through the oxidation of critical residues at active sites, enhancing the destructive reactions mediated by elastase [401].

In contrast, an overexpression of mRNA of SOD3 correlates with increased benign growth and is associated with apoptosis induction and the death of cancer cells [397]. The removal of O_2_^•−^ by SOD3 improves vasorelaxation by NO and enables NO to activate HIF-2α [402]. In turn, the HIF-2α expression in endothelial cells activates the WNT signaling pathway, where the components are involved in the stabilization and normalization of the endothelium, including the enhanced transcription of vascular endothelial cadherin (VED) [399]. This improves T cell extravasation from blood vessels into the TME and enhances anti-tumor activities [402]. An overexpression of SOD3 also attenuates the activity of MMPs and heparinase, which are involved in matrix degradation, angiogenesis, and metastasis [403].

### 4.8. Role of the Extracellular Matrix Components in the TME

Different factors contribute to remodeling of the extracellular matrix (ECM) in the TME. This remodeling is accompanied by an increased stiffness of the ECM and an increased synthesis and decomposition of the matrix components like collagens, hyaluronan, and others.

In healthy tissues, a high molecular weight hyaluronan is essential for maintaining tissue homeostasis. It exhibits anti-inflammatory and anti-proliferative properties and contributes to tumor resistance [404,405]. The fragmentation of high molecular weight hyaluronan by hyaluronidases or reactive species into low molecular weight polymers promotes tumor cell migration and proliferation as well as immune cell influx in the TME [406,407].

During the resolution of inflammation, the initiation of repair processes is indispensable for restoring homeostasis in the affected tissue regions. Under the action of TGF-β secreted from M2-type macrophages and some other cells, the components of the extracellular matrix (ECM) proteins, like collagen, are mainly produced by myofibroblasts during tissue repair [408,409]. At the same time, different TIMPs are also released by these cells to prevent matrix degradation by MMPs. Whereas in normal wound healing the generation of ECM proteins and TIMPs is transient and counter regulated by various MMPs that destroy excess matrix components, the accumulation of ECM components and scar formation dominates in fibrotic tissues.

An overexpression of MMP-2, MMP-3, MMP-9, and MMP-14 is associated with tumor metastasis [154,155]. MMP-2 and MMP-9 are involved in collagen IV degradation in the basement membrane [410]. MMP-14 contributes to collagen digestion around tumor cells and favors cell invasion and migration [411]. MMP-9 also exhibits gene modulatory functions upon binding to the α4β1 integrin in chronic lymphocytic leukemia [412].

MMPs are controlled by TIMPs. In different cancer types, TIMP-1 is upregulated and associated with a poor prognosis [413,414]. Increased levels of TIMP-1 are also associated with cachexia in patients with chronic pancreatitis and pancreatic cancer [415]. However, in another study, TIMP-1 was indeed associated with pancreatic cancer, but not with a significant weight loss [416]. Contrariwise, TIMP-3 suppresses tumor progression and promotes apoptosis induction [417,418]. Low levels of TIMP-3 increase the activity of MMPs in cancer [419]. Different cytokines and micro RNAs are involved in regulation of the expression of TIMP-1 and TIMP-3 [413,420].

Despite chemotherapy and surgical resection, numerous patients with triple-negative breast cancer developed a recurrence of the disease. In the TME, collagen IV has been identified by proteomic approaches as a major driver of enhanced cancer cell motility and metastasis after chemotherapy in both mice and patients with this kind of cancer [421]. An overexpression of collagen IV is known for pancreatic, gastric, colorectal, and breast cancers [422]. High circulating levels of collagen IV are associated with a poor clinical outcome [423,424]. In the basement membrane, the activity of the heme peroxidase peroxidasin is essential for cross-linking reactions between collagen IV fibrils [425,426]. This peroxidase promotes angiogenesis through extracellular signal-regulated kinases 1/2, protein kinase B, the focal adhesion kinase pathways, and the induction of several pro-angiogenic genes [427]. In invasive metastatic melanoma cells, a high expression of peroxidasin was reported [428]. Peroxidasin is upregulated in several kinds of cancers [429]. Abnormal collagen IV cross-linking by peroxidasin leads to an exceedingly dense ECM and a disordered matrix structure, as reported for lung tissue. In consequence, T cell infiltration is highly disturbed [430].

## 5. Conclusions

In tumorigenesis, different oxidant-based and protease-based cytotoxic agents are upregulated. Nevertheless, antagonizing principles are usually highly active to diminish the disastrous effects of these cytotoxic agents and to provide survival for tumors as well as tumor progression and metastasis.

In late-stage tumor cells, the enhanced formation of reactive species, increase in iron ions, and elevated values of free heme are antagonized by an overexpression of the numerous anti-oxidative protective systems. In addition, the pathways contributing to the maintenance of redox homeostasis are also upregulated in tumor cells despite existing in oxidative stress conditions. These counter regulations limit the cytotoxic activities of reactive species, metal ions, and free heme, protect tumor cells from the induction of cell death mechanisms, and prevent them from experiencing additional stress caused by therapeutic approaches like radio- and chemotherapy.

It is necessary to emphasize that an overexpression of these protective mechanisms was developed step by step during tumorigenesis and reached their maximum characteristic in late-stage tumors, especially in those tumors, which were malignant and prone to metastasis. As long as these antagonizing systems against oxidative stress are not fully developed, the probability exists to affect tumors successfully through different therapeutic options. Of course, scientists worldwide are trying to implement novel anti-tumor therapies based on the inhibition and/or downregulation of antioxidative mechanisms in tumors. Then, the cytotoxic potential of stress-related reactive species, metal ions, and free heme can better be applied to substantially damaged tumor cells.

Tumor cells affect their immediate neighborhood through the secretion of active agents, creating a pro-tumoral environment characterized by immunosuppression and an elevated production of ECM elements. During the resolution phase of inflammation, similar processes occur at inflammatory sites with the difference that they act only temporary but not permanently, as in tumors.

In the TME, the contribution of cytotoxic agents and their protective mechanisms is more complex than in tumor cells. In addition to reactive species and free heme, elastase, angiotensin II, and MMPs also exhibit elevated activities in comparison to healthy tissue areas. In the TME, cytotoxic agents are mainly derived from immune cells and stromal cells. Strategies against these cytotoxic agents include the upregulation or downregulation of protective principles, the depletion of essential metabolites, and the enhanced production of the ECM.

Superoxide anion radicals, hemoglobin released from leaky red blood cells, and free heme favor the depletion of NO and hinder the activation of cytotoxic T cells. These effects are elevated by low values of SOD3 and hemopexin as well as by the enhanced expression of arginases and the presence of hemoglobin released from leaky red blood cells.

Collagen and other ECM components form a barrier around tumor cells against invading immune cells. The increased production of the ECM is associated with tumor fibrosis. Otherwise, elastase, MMPs, and heparinase are involved in local matrix degradation that highly promotes tumor invasion and metastasis.

To sum up, despite an enormous knowledge about the interplay between cytotoxic agents and antagonizing principles in tumorigenesis, we are far from a thorough understanding how these mechanisms are embedded into the whole machinery of tumor-induced immunosuppression and metabolic alterations. Moreover, the individual expression of cytotoxic agents and protective mechanisms can vary considerably from patient to patient. It is a very great challenge to implement novel therapeutic approaches for individual cancer patients.

## Figures and Tables

**Figure 1 cancers-16-00081-f001:**
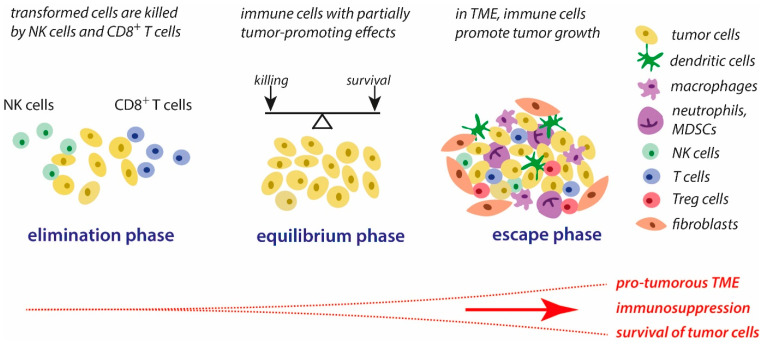
Concept of immunoediting in tumorigenesis. The three major phases of the interplay between tumor cells and immune cells are schematically presented. Further explanations are given in the text. Abbreviations: MDSCs—myeloid-derived suppressor cells, NK cells—natural killer cells, TME—tumor microenvironment, Tregs—regulatory T cells.

**Figure 2 cancers-16-00081-f002:**
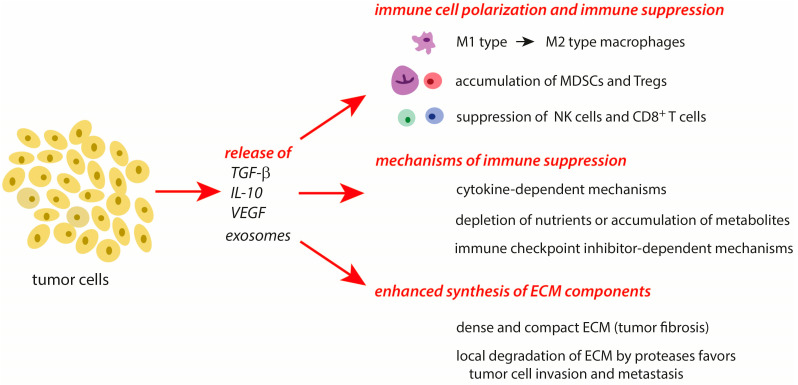
Major effects of tumor cells on the surrounding cells in the TME. The effects are mediated by the release of mediators from tumors cells and special conditions typical of tumors. Further explanations are given in the text. Abbreviations: IL-10—interleukin 10, MDSCs—myeloid-derived suppressor cells, NK cells—natural killer cells, TGF-β—transforming growth factor-β, Tregs—regulatory T cells, VEGF—vascular endothelial growth factor.

**Figure 3 cancers-16-00081-f003:**
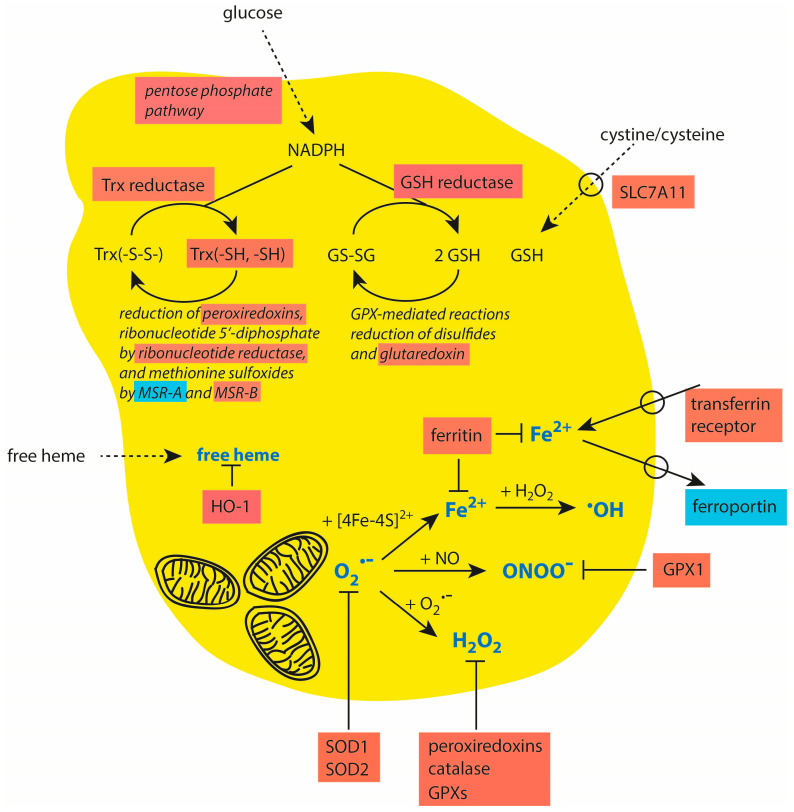
Major antagonizing mechanisms against the enhanced levels of reactive species, iron ions, and free heme in cancer cells. The upregulated enzymes, proteins, and pathways are displayed on red backgrounds. The downregulated mechanisms are given on blue background. For more details, see Section 3.3, Section 3.4, Section 3.5 and Section 3.6. Abbreviations: GPX—glutathione peroxidase, GSH—glutathione, GS-SG—oxidized glutathione, HO-1—heme oxygenase 1, MSR—methionine sulfoxide reductase, SLC7A11—cystine/glutamate anti-porter solute carrier family 7 member 11, SOD—superoxide dismutase, Trx—thioredoxin.

**Figure 4 cancers-16-00081-f004:**
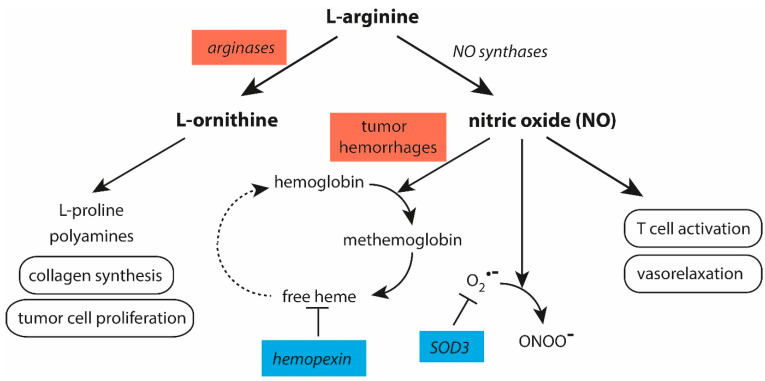
Factors contributing to the depletion of nitric oxide (NO) in the TME of advanced tumors. NO is generated by NO synthases from L-arginine. It is essential for the activation of CD8^+^ T cells and vasorelaxation. NO depletion is promoted by upregulated or enhanced factors (displayed on the red background) as well as by low-expressed proteins (presented on the blue background). Further explanations are given in the text. Abbreviations: SOD—superoxide dismutase.

**Table 2 cancers-16-00081-t002:** Effects of the proteins involved in iron and heme metabolism in cancer cells of late-stage cancers.

Antioxidative System	Functions	Effects in Cancer Cells	Cancer Types	References
Transferrin receptor	Iron uptake to cells	Upregulation	Liver, breast, colon, and lung cancer	[275,280]
Ferritin	Cellular iron storage protein	Upregulation	Esophageal adenocarcinoma, glioblastoma, and breast and colorectal cancer	[276,281]
Ferroportin	Iron export from cells	Downregulation	Prostate and breast cancers	[277,278,279]
Heme oxygenase 1	Removal of free heme	Upregulation	Multiple types of cancer	[282]

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
