# Peer review of "Inflammation-Associated Cytotoxic Agents in Tumorigenesis"

_cancers, 2023, doi:10.3390/cancers16010081_

Round 1

Reviewer 1 Report

Comments and Suggestions for Authors

This review illustrates the inflammatory processes associated with tumorigenesis and how cancer cells survive in the presence of inflammatory molecules. The author described very well the different mechanisms activated by tumors to protect themself from cytotoxic agents present in the tumor microenvironment. The review is accurate and complete.

I have only minor comments: 

- On page 8 in line 315, the bracket is missing;

- The following references are not written in the same format as the others: 127, 180-183, 222-235, 238-258, 320-328, and 335- 336;

- in the title of section 4.4 the full name of MPO should be written for better comprehension;

- in figure 2 I suggest adding exosomes as mediators released by tumor cells to underline their importance in cancer;

- Reading Table 1 is not immediate, maybe the defense systems reported should be separated by a line or more space;

- In Figure 3 the MSR-B up-regulation and MSR-A down-regulation, despite the color indication, are not adequately represented;

- the figure 4 is cut from the side.

Author Response

Thank you very much for your review and your helpful comments. The following changes have been performed according to your suggestion.

- On page 8 in line 315, the bracket is missing;

The missing bracket has been included.

- The following references are not written in the same format as the others: 127, 180-183, 222-235, 238-258, 320-328, and 335- 336;

The format is now identical for all references.

- in the title of section 4.4 the full name of MPO should be written for better comprehension;

The full name for MPO has been included.

- in figure 2 I suggest adding exosomes as mediators released by tumor cells to underline their importance in cancer;

In Figure 2, the word exosomes has been included.

- Reading Table 1 is not immediate, maybe the defense systems reported should be separated by a line or more space;

Table 1 is better readable now.

- In Figure 3 the MSR-B up-regulation and MSR-A down-regulation, despite the color indication, are not adequately represented;

In Figure 3, the assignment of thioredoxin-dependent mechanisms including MSR-A and MSR-B has been improved.

- the figure 4 is cut from the side.

This figure has been shifted to the left side.

Reviewer 2 Report

Comments and Suggestions for Authors

Overall, this manuscript is  very relevant and will be of great interest for the researchers in the field. Only minor comments were noted.

Line 84-88: the sentence should be reworded for improved clarity.

Line 115-116: the sentence starting with "in healthy tissues..." should be reworded for improved clarity.

Line 188: the author should provide additional information on exosomes.

Line 197: The author should briefly explain the role of the two types of MDSCs, if possible.

Line 317: The section 2.7 regarding the different pH of cancer cell, extracellular medium and TME is confusing. What is the difference between the TME and the extracellular medium of cancer cells? 

The words "cancer" cells seem to be use sometimes in place of tumor cell. A tumor cell is not necessary a cancer cell but could be either a cancer cell or a non-cancer cell, such as cells from the TME (immune cell, fibroblast, etc)

Line 578: The author list angiotensin II as protease based agents, this is mislead9ing and should be modified.

 Line 588: The sentence starting with ..."in TME..." should be reworded to improve its clarity.

Comments on the Quality of English Language

A few typos were noted in the manuscript. Some words should be replaced with more suitable ones to improve clarity of the manuscript.

Author Response

Thank you very much for your review and your helpful comments. The following changes have been performed according to your suggestion.

Line 84-88: the sentence should be reworded for improved clarity.

This sentence has been reworded.

Line 115-116: the sentence starting with "in healthy tissues..." should be reworded for improved clarity.

This sentence has been reworded.

Line 188: the author should provide additional information on exosomes.

A short explanation about the significance of exosomes has been included in this paragraph.

Line 197: The author should briefly explain the role of the two types of MDSCs, if possible.

A short explanation about differentiation of MDSC subtypes is included.

Line 317: The section 2.7 regarding the different pH of cancer cell, extracellular medium and TME is confusing. What is the difference between the TME and the extracellular medium of cancer cells? 

I only included “in normal cells” at the end of this sentence. I did not change the term extracellular pH, as pH values are compared between cancer cells and normal cells in this sentence.

The words "cancer" cells seem to be use sometimes in place of tumor cell. A tumor cell is not necessary a cancer cell but could be either a cancer cell or a non-cancer cell, such as cells from the TME (immune cell, fibroblast, etc)

At several places, especially in section 3, I replaced the term tumor cells by cancer cells.

Line 578: The author list angiotensin II as protease based agents, this is mislead9ing and should be modified.

I specified the role of angiotensin II.

 Line 588: The sentence starting with ..."in TME..." should be reworded to improve its clarity.

This sentence has been reworded.

Reviewer 3 Report

Comments and Suggestions for Authors

The author of the above manuscript has reviewed the latest knowledge about the altered balance between cytotoxic agents and protective mechanisms in tumorigenesis.

Overall, the review is well constructed and there is a substantial bibliographic reference that is appropriate for the aim and the topic of the review. I have few suggestions to the author before the acceptance for publication.

I would like to suggest the author to expand a bit the role of the immune checkpoint inhibitor PD-1- PD-L1 axis as it represents an important element for the so-called “cancer immune-evasion” mechanism.

A reference for the paragraph 1 in lane 69 is needed

I would like to suggest the author check typos in lanes 33,353, 653, 659, 806.

Figure 4 needs to be adjusted as it goes out of the margins. I would also adjust a bit the arrows in Figure 3

Author Response

Thank you very much for your review and your helpful comments. The following changes have been performed according to your suggestion.

I would like to suggest the author to expand a bit the role of the immune checkpoint inhibitor PD-1- PD-L1 axis as it represents an important element for the so-called “cancer immune-evasion” mechanism.

Some more information about the PD-1/PD-L1 axis is given in the revised version.

A reference for the paragraph 1 in lane 69 is needed

Two references are included.

I would like to suggest the author check typos in lanes 33,353, 653, 659, 806.

Typos have been corrected.

Figure 4 needs to be adjusted as it goes out of the margins. I would also adjust a bit the arrows in Figure 3

Figure is shifted to the left side. Slight corrections are performed in Figure 3.